# From Promises to Practice: Evaluating the Private Browsing Modes of Android Browser Apps

## ABSTRACT

Private browsing is a common feature of web browsers on desktop platforms. This feature protects the privacy of users browsing the Internet and, therefore, is widely welcomed by users. In recent years, with the popularity of smartphones, the private browsing mode has been introduced into mobile browsers. However, its deployment on mobile platforms has not been well evaluated. To bridge the gap, in this work, we systemically studied the private browsing modes of Android browser apps. Specifically, we proposed six private rules for mobile browsers to follow by combining the mobile browsing features with the previous research on private browsing. Furthermore, we designed an automated analysis framework, BroDroid, to detect whether mobile browsers violate these rules. Also, with BroDroid, we evaluated 49 popular browser apps crawled from Google Play. Finally, BroDroid successfully identified 58 violations, some of which come from the promised capabilities of the browser. We reported our discovered issues to the corresponding developers, and four of them (Yandex Browser, Mint Browser, Web Explorer, and Net Fast Web Browser) have acknowledged our findings. Our observation may be the tip of the iceberg, and more efforts should be put into improving the privacy protections of mobile browsers.

## CCS CONCEPTS

• **Security and privacy** → Software and application security.

## 1 INTRODUCTION

Private browsing is a common and popular feature of desktop browsers. This feature is designed to prevent any information related to browsing from being stored on the device being used [18]. A user study shows that 77% of non-technical participants use private browsing mode to protect their digital traces [26].

In recent years, with the popularity of smartphones, the private browsing mode has been introduced in mobile browsers, as shown in Figure 1. For example, the private mode of Chrome[1] is called the "Incognito Tab" and Edge calls it "InPrivate". In general, when switching to private mode, users are informed of prompting information, or the browser is switched to a dark background, indicating that browsing behaviors are being protected. Considering that the mobile phone plays an irreplaceable role in modern life, mobile browsers may have more opportunities to access user private data

---

[1]In this paper, except as otherwise noted, the mentioned browser is the mobile version.

*WWW 2024, May 13-17,2024, Singapore*
© 2023 Association for Computing Machinery.
ACM ISBN 978-x-xxxx-xxxx-x/YY/MM. . . $15.00
https://doi.org/10.1145/nnnnnnn.nnnnnnn

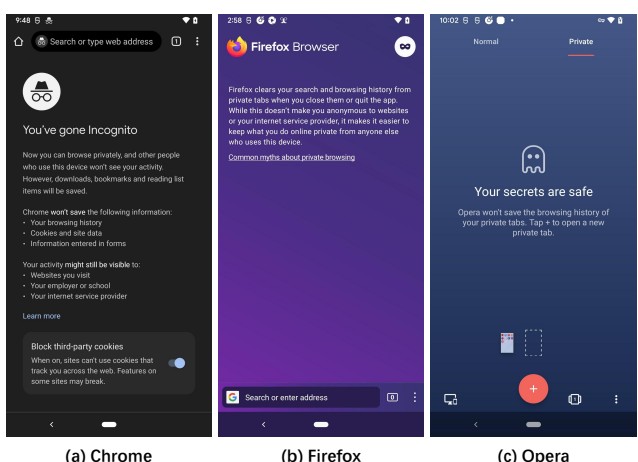

**Figure 1: Private browsing modes of browser apps.**

than PC browsers, such as passwords, visited websites, and search history. In addition, users are usually requested to provide their data when using a service on the Web, making it more possible for mobile phones to access user data. In 2016, the European Parliament introduced the General Data Protection Regulation (GDPR), which requires companies to protect user data with appropriate security measures [18]. Thus, mobile private browsing should provide a reliable service without leaving any trace of browsing activities.

The proper deployment of the private browsing mode on the PC platform has been widely discussed [11, 14, 18, 25, 30]. However, due to the enormous differences between PC and mobile platforms, PC browsers and mobile browsers face different kinds of adversaries, and the security mechanisms of PC private browsing can not be applied to mobile platforms directly. For example, the programs on the PC platform can view the files held by other programs without restriction. Instead, apps cannot view each other's files on Android by default due to the app isolation design. Therefore, even though both PC private browsing and mobile private browsing both aim to clear browsing information, they should consider different technical details. On the other hand, most previous studies are manual analyses rather than automatic work and only investigated a minority of specific PC browsers. As a result, the status of private browsing deployment in mobile browsers is not well understood.

**Our work.** In this study, we conducted a systematic analysis of mobile private browsing and focused on the Android platform due to its high market occupancy on mobile platforms. Specifically, we proposed six private browsing rules for mobile browsers to follow by collecting the private browsing features claimed by the browsers themselves (in the Appendix) and referring to previous related works about PC browsers. Additionally, after solving a series of technical challenges (such as UI positioning), we designed

an automated analysis framework, BRODROID, to detect whether mobile browsers violate these rules.

To give a complete view of private browsing deployments on mobile platforms in the wild, we conducted a real-world evaluation on 49 popular Android browsers with over 5M+ installations. In total, we identified 58 violations, meaning that every browser violated one rule on average, even though some of them claimed that their private browsing guaranteed the corresponding capabilities (3/49, 6.1%). That is, the deployment of private browsing in mobile browsers still needs to be improved.

Our work not only reflects the general state of mobile private browsing implementation but also indicates that the awareness of mobile private browsing needs to be improved. In this case, our work can benefit developers and communities as it systematically evaluates mobile private browsing.

**Responsible disclosure.** We reported our discovered issues to the corresponding developers, and four of them (Yandex Browser, Mint Browser, Web Explorer[2], and Net Fast Web Browser) have acknowledged our findings.

**Contributions.** Here, we list the main contributions of this paper.
- *Systematic study.* We systematically studied the implementations of private browsing on the Android platform. Also, we proposed six rules for mobile browsers to follow.
- *Analysis tool.* We designed an automated analysis framework – BRODROID, which can evaluate the performance of private browsing in Android browsers based on our predefined rules.
- *Real-world measurement.* With BRODROID, we evaluated 49 popular browser apps and identified 58 violations. Our study shows that private browsing of mobile apps is not well implemented, even with some of the promised features.

## 2 BACKGROUND AND MOTIVATION

This section covers the Android security mechanisms and private browsing background. In addition, we also give a real-world example of the wrong implementation of private browsing.

### 2.1 Android Security Mechanisms

Android operating system is a multi-user Linux system where each app is a different user. It implements the principle of least privilege [20]. That is, an app should never be assigned more privileges than it needs. Its core security mechanisms are listed below.

**App sandbox.** Sandbox is an isolated mechanism that promises an app can not affect other apps outside its boundaries [15]. As the Android platform employs Linux user-based protection, it uses the user ID to set up a kernel-level sandbox to isolate apps from each other [5]. Also, the sandbox guarantees that the app running in it will not impact resources outside, like the file system and network. It effectively prohibits apps from breaking the resource another app uses or causing a data leak.

**Permission.** User private data and sensitive system resources are protected by the permission mechanism. By default, each app runs in a process with a low-privilege user ID, and apps can access only their own files [17]. If apps want to access system resources, they must apply for corresponding permissions. For example, system

---

[2]Package name is com.explore.web.browser.

state and user contact information are restricted for app access due to the permission mechanism [7].

**Private data storage.** According to the Android developer documentation [21], it is recommended that all private user data be stored in internal storage. An app can access its internal files by default, and others cannot access them (differently from PC platforms) [22]. When an app is uninstalled, the OS will delete its corresponding files saved within internal storage.

### 2.2 Private Browsing

Private browsing is a feature that makes the browser not save browsing history, cookies, and other user data when a user browses the websites, regardless of the network protocol (e.g., HTTP and HTTPS). In addition, others who can access that device cannot see its user's activities while activating the private browsing mode. As the Android has its own web architecture and security mechanisms (e,g., data storage difference), the private browsing techniques implemented on PC platforms cannot be migrated to mobile platforms directly. Besides, these security mechanisms can protect only against app-level adversaries, such as malicious apps. Based on previous studies [11, 36], we propose a new model that considers attacks not from the perspective of the app, but from the web and local adversaries.

**Threat model.** A private browser should not leave any trace of users' browsing activities. Although browser apps are protected by Android security mechanisms, powerful non-app adversaries can still obtain users' private browsing activities. Here, we consider two kinds of adversaries:
- *Local adversary:* A local adversary can physically access a user's mobile phone. From the perspective of digital forensics, these adversaries can be so powerful that they can get the root privilege through popular tools such as Magisk [33], KingRoot [27], and Xposed [32]. Then, the data generated by all apps can be viewed by them. The local adversary aims to uncover the user's activities during private browsing.
- *Web adversary:* A Web adversary can control websites that the user visits [11]. The adversary can view all the content of the network traffic between the user and the website. The target of the Web adversary is to link a user in private mode to the same user in normal mode.

### 2.3 Motivation

The primary requirement of private browsing on any device is not to save browsing history (website addresses). However, for the mobile platform (we mainly discussed the Android platform due to its high occupancy in the mobile market), related files generated by Web browsing are mainly stored in the browser app's internal storage after a user ends his private browsing activities, which is protected by its security mechanisms. Thus, to roundly figure out whether a browser app saves private browsing history, it is necessary to examine its internal storage content.

We tested several popular web browsers with private browsing mode and discovered that some did not live up to their promise of providing complete privacy. For example, *Yandex Browser*, which has 100M+ downloads on Google Play, claimed, "Your history, searches, and passwords will not be saved" on its private tab page

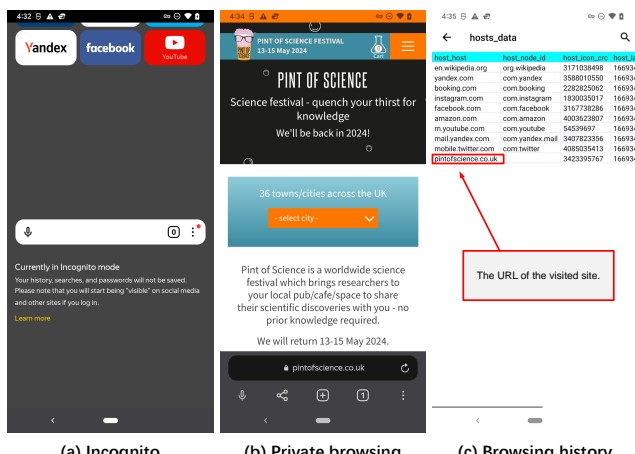

(a) Incognito          (b) Private browsing          (c) Browsing history

**Figure 2: Private browsing of Yandex Browser.**

(viewed as Figure 2 (a)). To verify its claims, we first opened a new private tab, and in this tab, we visited a website that the browser had never visited before, and its address is "https://pintofscience.co.uk/" as Figure 2 (b) shows (as previous work on PC browsers [23] did), then we stayed at this webpage for a while (around five seconds). After that, we closed this private tab and continuously tapped the "back" button of the phone to exit the browser. After 30 seconds, we inspected the internal storage of the app and found that the website we visited was stored in the database file "com.yandex.browser/databases/dashboard_service", and the concrete table name is "hosts_data" (viewed as Figure 2 (c)).

It is important to ensure that mobile browsers can achieve their private browsing goals, as mobile browsing is more likely to access users' private data. Therefore, it is crucial to examine the implementation of mobile private browsing and evaluate its effectiveness.

## 3 PRIVATE RULES FOR MOBILE BROWSERS

We proposed six rules to systematically evaluate whether the browser effectively implements private browsing against the two adversaries listed above. In particular, we proposed these rules based on two considerations: (1) the private browsing features of PC browsers discussed in previous works; (2) the private browsing capabilities promised by the app developers (as shown in Figure 1).

Generally, after a user finishes Web browsing (closes private tabs, private mode, and the browser app), browsers should satisfy the following rules to prevent threats from a *local adversary*:

- **Rule 1**: *Do not save browsing history.* Browsing history leaks the addresses of websites users visited, and it can directly reveal the user's browsing activity [10, 11, 25, 26, 30, 36, 37].
- **Rule 2**: *Do not save cookies.* The purpose of cookies is to add a state to the HTTP protocol so that the server can retrieve the previous user. It can not be saved after private browsing to avoid leakage of user network trace [10, 11, 25, 30, 36, 36, 37].
- **Rule 3**: *Do not save the Web cache.* Web cache can reduce the overall network delay by storing files or data on the client side. For example, the Web cache file may change size after a user browses a picture online. Furthermore, the modified part stores

the data of that picture (maybe not the complete picture). Since Web cache can be sensitive to uncovering a user's browsing activity, it should not be saved [10, 11, 25, 26, 30, 36, 37].
- **Rule 4**: *Do not save forms.* Forms are a common tool for submitting user data, such as user names and passwords. As form data submitted in private browsing should not be exposed to the adversary, browsers should not save it [11, 36, 37].

Besides, when a user visits websites, to resist the threat from a *Web adversary*, private browsing should follow the rules below:

- **Rule 5**: *Block third-party cookies.* A third-party cookie is a cookie set by a website other than the one a user is currently on. It can help third parties create a user profile by collecting a staggering number of private data, such as the user's IP address and other device details. Therefore, a third-party cookie should be blocked to prevent leakage of user identification information [25].
- **Rule 6**: *Do not share cookies in different modes.* This rule prevents a website from correlating a user in a normal session to the same user in a private session. For example, browsers should promise that after a user logins to a website using the username and password in normal mode, the user should enter this username and password again to login to the same website in private mode later instead of directly getting to the web page in login state [25].

Notably, these rules have been widely discussed and accepted [10, 11, 25, 26, 30, 36, 37] and all browsers should follow them.

## 4 DESIGN OF BRODROID

To detect whether browser apps follow these rules, we designed an automated analysis tool – BRODROID. Here, we describe its detailed design.

At a high level, BRODROID treats the private browsing implementations of the browser apps as black boxes. It checks whether browsers achieve their private browsing goals. Specifically, BRODROID takes browser apps as input and outputs a report of the detection results corresponding to the pre-defined rules. Figure 3 illustrates the overview workflow of BRODROID, which consists of four modules, including environment preparation, browsing automation, network traffic analysis, and local storage analysis.

(1) *Environment Preparation (§4.1).* First, browsers should be installed on a test device that already got the root permission. BRODROID extract their metadata for further use. Additionally, two test sites should be configured correctly to provide a test environment for the proposed rules.
(2) *Browsing Automation (§4.2).* Then, the tested browser is automatically launched and visits our self-built test websites in normal and private modes driven by BRODROID based on Appium [2].
(3) *Network Traffic Analysis (§4.3).* Next, BRODROID catches network traffic throughout the normal browsing and private browsing process using tcpdump [8]. Traffic packets whose source IP and destination IP are related to our test website are focused and analyzed (for **Rule 2** and **5-6**).
(4) *Local Storage Analysis (§4.4).* BRODROID analyzes the stored files related to the browser after completing normal and private browsing, respectively, to confirm the private data storage status (for **Rule 1-5**).

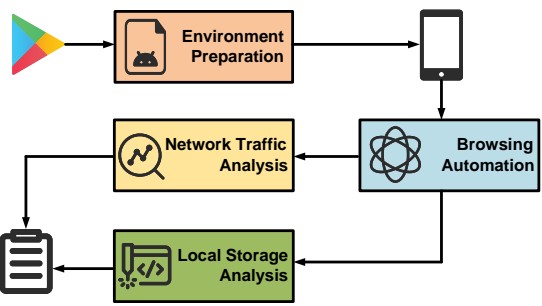

**Figure 3: Overview of BroDroid.**

## 4.1 Environment Preparation

We crawl popular browser apps for BroDroid as input. BroDroid extracts basic information from `AndroidManifest.xml` files of apps using the Androguard [1], including launcher activity, package name, and version number. Then, in the following steps, BroDroid uses this information to launch browsers.

**Test websites.** To validate the rules, we set up two websites in the LAN for browsers visiting. As shown in Figure 4, the test website (`10.102.32.216`) provides a login page where users can input their username and password in the input boxes. The form data will be posted to the test website once clicking the "submit" button. Furthermore, we set two pairs of username and password for the test website to distinguish whether a browser is in normal mode or private mode: "try" for normal mode and "uvw" for private mode. The username and password are the same. Additionally, to eliminate the interference of normal browsing and make our test environment closer to the user's browsing behavior, we introduce a new webpage from a general external website for private browsing.

In normal mode, the test website returns an HTML file only containing a "success" text to the browser after logging in. At the same time, the test website sets the cookie "`Flag=1`" for this session and returns it to the browser. In private mode, the test website returns an HTML file containing an "img" tag except for the "success" text. Similarly, the test website sets the cookie "`FlagPrivate=2`". In addition, this "img" tag has an attribution "src" of which value is the resource path of another test website (`10.102.32.217`). It makes the browser request an image file from this third-party website. When receiving the request, it sends the image back to the browser and sets a third-party cookie "`FlagThirdParty=3`" along with it.

## 4.2 Browsing Automation

To meet the needs of large-scale analysis for each browser to verify whether it complies with the above rules, we designed an automated browsing process, as shown in Figure 4. It simulates the user's everyday browsing operations and covers all our proposed rules. Mainly, it can be divided into the following four operations:

- **Ops 1:** In normal mode, BroDroid taps the search bar and enters the test website address to access it. And then, it enters a username and password on the login page and clicks the "submit" button to post these data. Then, BroDroid drives the browser to exit itself by continuously sending the "back" command to get to the phone's homepage.

- **Ops 2:** BroDroid repeats the previous operation as the former step but takes a screenshot when inputting the username.
- **Ops 3:** BroDroid first switches to private mode and opens the same website browsed in normal mode. Subsequently, BroDroid enters a different username and password and taps the "submit" button. Then, BroDroid visits a new webpage that does not occur in normal mode. After a while, BroDroid closes the browser.
- **Ops 4:** BroDroid repeats the same operation as Ops 3 except for visiting the new webpage. In addition, it takes a screenshot when inputting the username.

**Challenges.** We encountered two challenges in implementing browsing automation. In particular, to successfully visit test websites in private mode automatically, we need to position the UI components in the browser to accomplish at least two tasks: (1) launch of private browsing mode, (2) access to the test website. However, the UI layouts of browser apps are diverse, which creates an enormous barrier to implementing automated browsing.

For Task (1), some browsers can launch their private mode by the corresponding launching activities, like the `IncognitoTabLauncher` activity for Chrome. However, other browsers cannot launch private mode in this way because the launching activity is restricted by its attribute (`exported="false"`), or there is no activity for browsers to launch this mode at all. Therefore, BroDroid needs to click the private mode switch to open it, as users do. This operation is not easy because it is unknown whether the private switch button is displayed on the browser's homepage.

For Task (2), when we tried to drive the browser to visit our test websites, we found that many browser apps prevent their private windows from being viewed by the tool we used (i.e., Appium), directly affecting the UI positioning.

**Solutions.** To overcome the challenge in Task (1), we first investigated the most popular browser apps to find the commonality of switching to private mode. Then we proposed empirical solutions for an automated process to follow, summarized as follows:

- The private mode can be launched in one step: users can tap the switching button on the home page of the browser, and then it switches to private browsing mode at once, e.g., Figure 1(b).
- The private mode can be launched in two steps: first, the user taps the button next to the search bar in the top right corner of the home page, and then it shows the indicated text, like "New Incognito Tab". Second, the user goes on to tap it, and a new private tab starts, e.g., Figure 1(a).
- The private mode can be launched in three steps: first, the user taps the button that means "tabs view" of the bottom toolbar, and then more function buttons emerge. Second, the user taps the text of the private hint so that it can switch to private mode. At last, the user taps the plus sign (presented as "+" in almost situation) to open the private browsing tab, then the user can browse in private mode, e.g., Figure 1(c).

To solve the challenge of Task (2), we studied the reason why Appium cannot view the layout, and then we found that it is because of apps' `FLAG_SECURE` employment. To block this feature, we utilized the LSPosed framework [6], an ART (Android Runtime) hooking framework, and rebooted the test device to enable it. After solving these challenges, we can finally position the UI components to perform Tasks (1) and (2).

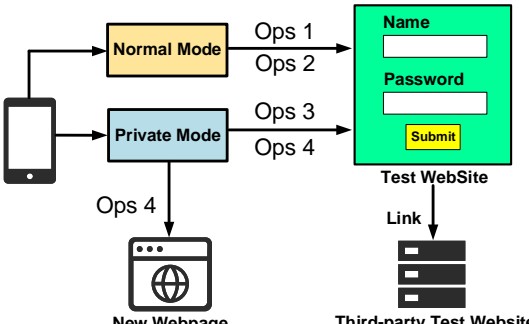

**Figure 4: Automated browsing process.**

## 4.3 Network Traffic Analysis

Network traffic analysis mainly detects whether browser apps comply with **Rule 2**, **5**, and **6**. BRODROID uses Tshark [9] to analyze traffic packets associated with the test sites.

**Normal mode traffic.** BRODROID captures the network packets in Ops 2. And BRODROID detects **Rule 2** in normal browsing according to these network traffic packets.

- For **Rule 2**, in Ops 1, after login, the test website sets a cookie for the browser (i.e., "Flag=1"). If the browser saves the cookie, it may be carried to access the test website the second time of normal browsing. If "Flag=1" is found, it violates **Rule 2**.

**Private mode traffic.** Different from the normal mode traffic analysis, BRODROID captures the network packets twice in private browsing of Ops 3 and Ops 4, respectively. And we detect **Rule 2**, **5**, and **6** there:

- For **Rule 2**, like normal mode traffic analysis, BRODROID detects whether the cookie got from the first private browsing (i.e., "FlagPrivate =2") in the traffic packets of Ops 4.
- For **Rule 5**, when the browser loads images from a third-party website in Ops 3, the response message will carry a third-party cookie ("FlagThirdParty=3"). BRODROID searches for that cookie in the package of Ops 4 to verify if it was saved and carried.
- For **Rule 6**, BRODROID searches for the cookie key-value pair ("Flag=1") in traffic packets of Ops 3 to analyze whether the browser shares cookies received in normal mode with private mode.

Overall, for **Rule 2**, **5**, and **6**, if those key-value pairs are found, the browser violates the corresponding rules.

## 4.4 Local Storage Analysis

Local storage analysis mainly checks whether the browser apps comply with **Rule 1 - 5**. BRODROID double-checks **Rule 2** and **5** using local storage analysis to support the accuracy of our research method. We manually analyzed some cases to find the storage location of user private data. The results indicate the app database records some browsing information, and some files related to the Web cache are written into the "cache" directory. Thus, we take database files, "cache" directories into consideration to analyze local storage. Specifically, BRODROID analyzes the internal storage of the browser after the end of normal browsing and private browsing.

**Database content analysis.** User-submitted form data, browsing history, and cookies are usually stored in database files. For **Rule 1**, **2**, **4**, and **5**, BRODROID needs to verify the following data in database files to judge the compliance of the rules in private browsing:

- For **Rule 1**, we need to check if the URL of the new webpage entered in Ops 3 exists in the database files.
- For **Rule 2**, we need to check if the cookie got from private browsing (i.e., "FlagPrivate=2") is in the database files.
- For **Rule 4**, we need to check if the username and password entered in Ops 4 (i.e., "uvw") exist in the database files.
- For **Rule 5**, we need to check if the third party's cookie ("Flag-ThirdParty=3") exists in the database files.

The browser app violates the corresponding rule if the above data is found.

**Differential analysis.** To verify **Rule 3**, BRODROID analyzes the difference between the app storage after normal browsing and private browsing. In detail, for database files and cache directories, it compares their file size of normal browsing and private browsing. As the image in Ops 3 - 4 is around 1.8 MB, it can make an apparent change in file size if it is not cleared in private browsing. It is taken as the Web cache representative. Empirically, if a file size changes over 1 MB, it is thought to violate **Rule 3**.

**Screenshot analysis.** To further verify **Rule 4**, a screenshot in Ops 4 is used to check whether the username in Ops 3 (i.e., "uvw") is auto-filled or prompted in a drop-down box near the input box or not. If the full username is identified by character recognition, it means that the last private browsing saves the form data. Therefore, the browser disobeys **Rule 4**.

In addition, to distinguish private browsing features, we apply the same local storage analysis method to detect **Rule 2**, **4** in normal browsing, with different key-values "Flag=1" and "try", respectively.

## 5 EVALUATION RESULTS AND FINDINGS

This section describes how the experiment was conducted and summarizes our findings.

## 5.1 Experiment Setup

**Implementation of** BRODROID. We implemented a prototype of BRODROID. It was built on Appium, tcpdump, and Tshark. In total, we implemented it with 2,460 lines of Python code.

**Browser app dataset.** We collected 60 browser apps on Google Play with over 5 million downloads. Among them, four apps not equipped with private browsing mode were filtered out. Besides, we filtered out two browsers that could not run on our test device (these browsers crashed at runtime). Finally, the remaining 54 browser apps make up our dataset.

**Execution environment.** We took a rooted Pixel 3a mobile phone with Android 12, as our test device. It was connected to the PC using a USB interface and turned on the "development option" and "debugging option". Also, in our experiment, BRODROID runs on a Windows 11 PC equipped with Intel Core i7 2.50GHz CPU and 16G RAM. Moreover, our 2 test websites are deployed on 2 Apache servers, which operating system is Ubuntu 20.04, equipped with Intel Xeon Gold 6226R CPU 2.90GHz and 256G RAM.

**Table 1: Overall detection results.**

| Violated Rules | Rule 1 | Rule 2 | Rule 3 | Rule 4 | Rule 5 | Rule 6 |
|---|---|---|---|---|---|---|
| # of browser apps | 3 | 17 | 15 | 3 | 4 | 16 |
| % of browser apps | 6.1% | 34.7% | 30.6% | 6.1% | 8.2% | 32.7% |
| # of promised capabilities | 35 | 11 | 12 | 12 | 11 | 2 |
| # of violated capabilities | 1 | 2 | 2 | 0 | 0 | 2 |

We installed the listed browser apps on the phone and configured them, including accepting user privacy and service policies. After that, the browser apps kept their default configurations.

## 5.2 Findings

Our findings are summarized in Table 1 and Table 3 (in the Appendix). At the app level, BRODROID successfully evaluated 49 browser apps[3] and discovered 58 violations. At the rule level, few browsers violate the rules widely known as classic private features: three browsers save the browsing history (**Rule 1**, 6.1%)), and three browsers save form data (**Rule 4**, 6.1%). Furthermore, some browsers do not perform web-related behaviors well: Four browsers save third-party cookies (**Rule 5**, 8.2%), and 16 browsers share cookies with private mode (**Rule 6**, 32.7%). Besides, 17 browsers save the cookie (**Rule 2**, 34.7%) in private browsing, and 15 browser apps save the web cache (**Rule 3**, 30.6%). These violations can disclose user identity and privacy. At the promised capabilities level, these 49 browsers claimed 86 capacities corresponding to our rules, and 6 of these capabilities were violated (from 3 browsers, respectively). Table 1 shows the statistical result of our evaluation, indicating that mobile private browsing is not quite reliable. The causes of these problems are various. On the one hand, developers need more systematic consensus on mobile private browsing, although the same question has been well-discussed on PC platforms [11, 14, 18, 25, 30]. On the other hand, even though some developers pay attention to enhancing private browsing by means, they may ignore testing their products or private browsing effectiveness. The details of our analysis results are as follows:

**To Rule 1.** Three browser apps (i.e., Yandex Browser, Maxthon, Samsung Internet Beta) save the private browsing history. Generally, no browsing history is the most basic requirement of private browsing. However, some browsers still leave the related information, though not intentionally. In our experiment, the browsers do not save the URL of the new webpage's on purpose, but they may record the browsing address when describing other data. For instance, Maxthon recorded the browsing history in the database file `com.mx.browser/databases/mxcommon.db`, and the table name is "mxfavicons". According to this name, we can infer that this table is used to store the favorite icon information of websites. This table has three attributes (or column names): host, favicon, and `TOUCH_ICON`. We found that the URL is listed in the "host" column. This violation reveals that browsers do not clear the browsing history as fine-grained.

**To Rule 2.** We used two methods to verify this rule, and 17 browser apps failed to pass both. Besides, we observed an interesting fact contrary to our experience. Ordinarily, in private browsing, a cookie should be temporarily saved before a current session ends and

[3]Five browser apps cannot access our test sites. We give the reasons in Section 6.

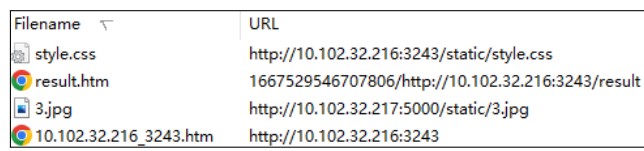

**Figure 5: Cache files of Aloha (Rule 3).**

deleted along with this session. However, we find Maxthon does not save the private cookie in local storage immediately after successful login. In contrast, if staying on the web page for a while, it will do so. In fact, considering the data restoration threat, it is recommended that browsers not save cookies in private browsing rather than deleting them later. In addition, for the privacy browser DuckDuckGo, it is needed to be clicked a specific button on the toolbar before closing the browser to delete data generated by browsing; otherwise, it will violate **Rule 2**. However, it requires the user to pay more attention to cleaning their browsing data, which may worsen the user experience.

**To Rule 3.** For cache file directories, 15 browser apps change their cache directory size to over 1 MB after private browsing. In contrast, no database files of these apps change so much. Taking Aloha as an example, to ensure the correctness of the verification results, we reinstalled this browser app and visited the test website in private mode. After successful login, we closed the private tab, private mode, and browser. According to the recorded file paths (i.e., `data/data/com.aloha.browser/cache/WebView/Default/HTTP Cache`) of our former detection results, we inspected this cache file using a tool called ChromeCacheView [3]. It showed that some cache files directly reveal what the user browsed in private mode. As Figure 5 shows, the cache files of private browsing include the figure (i.e., 3.jpg) from our third-party test website and an HTML file named "result.html" that can only be returned in private login. This finding means the browser did not remove the web cache generated in private browsing. Besides, while inspecting the local storage cache files of these 15 browser apps, we found that some HTML files and images from the new webpage used to verify **Rule 1** also existed. The HTML file name contains the exact website address. As this address information was stored in the web cache form, we considered saving the web cache instead of browsing history.

**To Rule 4.** Three browser apps (i.e., XBrowser, UC Browser, Seznam.cz) saved the form data. For XBrowser and UC Browser, it can be observed that the login page of our test website preloads the username we input last time, when reopening the browser and logging to the test website. For UC Browser and Seznam.cz, these form data were stored in database files. However, we searched for the username and password text in the local storage of XBrowser but failed to find them. Therefore, we considered its form data was stored as Web cache somewhere.

**To Rule 5.** Our result shows that four browser apps (APUS, Maxthon, XBrowser, Seznam.cn) save the third-party cookie. We detected this rule using two kinds of methods: network traffic analysis and local storage analysis, and none of these browsers can pass either of these methods. Moreover, these results are confirmed in two ways: searching the database files to look for the third-party cookie and printing the cookie on the third-party test website.

Table 2: Detection results of normal browsing.

| Violated Rule | Rule 2 | Rule 4 |
|---|---|---|
| # of browser app | 45 | 22 |
| % of browser app | 97.8% | 47.8% |

We find all these browsers saved the third-party cookie in their database file, and the concrete file path is `data/data/packagename/app_webview/Default/Cookies`, and the table name is "cookie". Besides, these browsers all carried that cookie when asking for the image again, e.g., Maxthon, showing as Listing 1.

```
1  Hypertext Transfer Protocol
2  GET /static/3.jpg HTTP/1.1\r\n
3  Host: 10.102.32.217:5000\r\n
4  ...
5  X-Requested-With: com.mx.browser\r\n
6  Referer: http://10.102.32.216:3243/\r\n
7  ...
8  Cookie: FlagThirdParty=3\r\n
9  ...
10 [Full request URI: http://10.102.32.217:5000/static
      /3.jpg]
```

Listing 1: HTTP message with third-party cookie (Rule 5).

However, it is easy to distinguish and block it because the third-party cookie has a different domain from the first-party cookie that roots in the website the user is visiting. In particular, some browsers, like Chrome, remind the user that blocking third-party cookies is the default setting when opening a private tab.

**To Rule 6.** In total, 16 browser apps share cookies in normal mode with private mode. Taking APUS Browser as an example, when it visited the test website in the first private browsing, one package was captured as Listing 2 shows. In our setting, the cookie "Flag=1" can only be obtained in normal browsing. However, if the private mode allows this cookie to be shared, it may inadvertently provide key information for websites to link a user in private mode with the same user in normal mode.

```
1  Hypertext Transfer Protocol
2  Host: 10.102.32.216:3243\r\n
3  ...
4  X-Requested-With: com.apusapps.browser\r\n
5  ...
6  Cookie: Flag=1\r\n
7  ...
8  [Full request URI: http://10.102.32.216:3243/]
```

Listing 2: HTTP message with shared cookie (Rule 6).

**Normal browsing feature.** As we stated before, we also detected **Rules 2** and **4** for 46 browser apps' normal browsing (three privacy browser apps are ruled out). The results are shown in Table 2. Comparing the normal mode with the private mode, it is evident that the latter does pay attention to protecting user privacy. Accordingly, many browsers do not save form data even in normal browsing. It makes sense that form data is directly related to users' privacy and should better not be saved.

**Chromium-based browsers.** Chromium [4] is a well-known open-source browser project. Based on the Chromium developer documentation and our assessment of the essential components of the browsers, we identified a total of 23 browsers that are based on Chromium. These browsers are listed in Table 3. In fact, of these 23

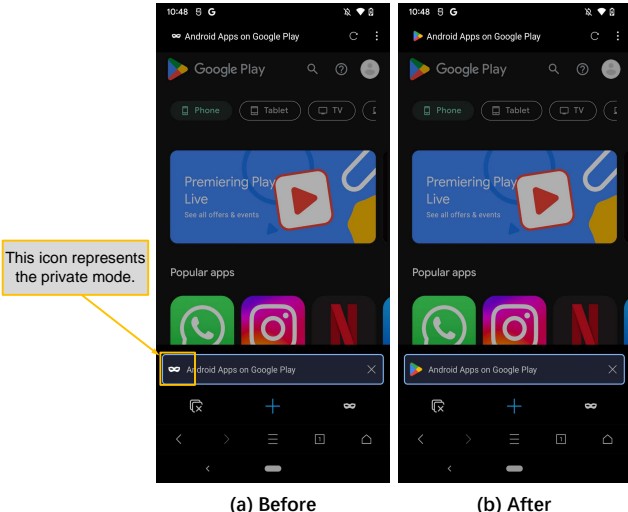

(a) Before      (b) After

This icon represents the private mode.

Figure 6: The private mode changes to normal mode when reopening Xbrowser.

browsers, 21 did not violate any rules, including the Chrome family of products (3 browsers). However, two browsers (i.e., Yandex Browser and UC Browser) violated one rule, respectively. Therefore, it is inferred that these two Chromium-based browsers present an unexpected threat to private browsing when customizing their products. On the whole, Chromium-based browsers behaved better than other browsers. Indeed, the private browsing of non-Chromium-based browsers is not necessarily flawed. However, achieving right-on private browsing is not easy.

## 5.3 Accuracy

**False positives.** We find three false positive cases in 58 violations about Brave, Ecosia and Mi Browser) from **Rule 3**. Based on our detection reports, three browsers actually changed the size of recorded cache files to over 1 MB. However, we cannot find the test image in those files. Also, those cache files cannot be parsed by Chrome-CacheView. Therefore, we cannot find what kind of files caused its size change. Finally, we determine them as false positives. In addition, these three browsers do not violate any of the six rules.

**False negatives.** We found one false negative during our manual verification. It was about UC Browser from **Rule 6**. we find UC Browser does not save any cookies after closing itself. Thus, it failed to verify **Rule 6** due to the disappearance of the cookie in normal browsing (i.e., Flag=1), and BRODROID incorrectly considered it complied with this rule because this cookie was not found in Ops 3. However, if we get this cookie in normal mode and switch to private mode without closing the browser, it will share this cookie with a new private tab. Therefore, it is considered a false negative. In addition, we found that the storage of private browsing data is related to time for some browsers. That is, the time to analyze storage after closing private browsing can affect the evaluation results. In our experiment, we stayed 30 seconds after closing browsers to leave enough time for browsers to clear their browsing data.

After that, we analyzed the storage status. If the stay time is shorter or there is no remaining stay time, the browsing data may not be completely cleared.

## 5.4 Case Study

To demonstrate how an inappropriate implementation of private browsing exposes user privacy, we take Xbrowser as a case. Xbrowser provides two ways to enter private browsing: one is to turn on the privacy mode switch, and the other is to create a new private tab. The former switches all tabs in the browser to private mode (both existing and new ones), while the latter switches only new tabs to private mode. When the private mode switch is used to enter the private browsing state, the contents of the existing tabs are retained. Then, when the switch is turned off, the contents of the private tabs are retained and converted to normal contents. If the user exits the browser at this point, the previous privacy data will not be automatically cleared. Additionally, no matter how private browsing is accessed, if the user closes the browser directly through background management (a common behavior), all privacy content will be loaded when the browser is relaunched. In contrast, if the user exits through the browser's exit button, the browser will clear the data normally. Figure 6 shows the screenshots taken while testing Xbrowser. We first opened a website in a private tab, and in the screenshot, we can see that the page is in private mode; then, we exited the browser without closing the current tab and reopening it. It can be found that the web page just reloaded, and the tabs changed to regular tabs.

We believe that browser app developers should not require users to actively turn off privacy mode before exiting the browser. It requests browser developers to pay more attention to the logic of their browser's private mode implementation and develop products from the user's perspective.

## 6 DISCUSSIONS

There are several potential threats to the validity of our study:

*Positioning UI elements.* Due to significant differences in UI design among browser apps, BRODROID may not position some UI elements. For example, the privacy toggle button of UC browser is a mask icon, and this mask icon does not contain any text information or resource-id description, BRODROID cannot handle this situation. There are a total of four browser apps, and BRODROID evaluated these browser apps with the help of manual work.

*VPN/Tor-based browsers.* Four browser apps use VPN services or Tor network [16] for access to the Internet by default, and their proxy nodes cannot access our LAN IP addresses. Since there will be an uncertain delay when accessing public IP addresses, for the sake of test efficiency, we did not host the test website on the public network. In addition, the VPN/Tor connections of these browser apps are not stable in our region, so we did not evaluate them during the experiment.

*Non-uniform privacy data cleaning conditions.* There is no uniform standard for triggering browser apps to clean the data generated in private modes, of which may be closing the browser apps, closing private tabs, or exiting the private mode. During our browsing automation process, simply closing the browser apps does not guarantee that they will close the private tabs and exit the private mode.

In addition, there is no uniform method of closing the browser apps along with closing the private tabs and private mode automatically. Therefore, we manually close browser apps, close private tabs, and exit private mode (if an app cannot do this work by our automation operations) to trigger the data cleaning condition.

## 7 RELATED WORK

**Private browsing analysis.** Wu et al. [37] compared four desktop and mobile browsers (Chrome, Firefox, Safari, Edge) to study their private browsing strategies, revealing differential implementations on both platforms. Aggarwal et al. [11] analyzed the private mode implementations of several popular desktop browsers and put forward the goals that private mode should achieve. Subsequently, the works of Lerner et al. [29] and Zhao et al. [39] analyzed the privacy breach issues caused by third-party browsers' plugins and proposed the corresponding identification and improvement approaches. Wu et al. [36] showed that the browser's explanation about private browsing should be clearer to reduce the user's misunderstanding. There is also some work focusing on the design and implementation of privacy browsing frameworks, such as Veil [35], PrivateDroid [28] and CYCLOSA [31]. Mobile private browsing has been relatively overlooked compared to its PC counterpart. Our work focused on mobile private browsing and evaluated large scale browsers, rather than a few specific popular browsers.

**Forensic investigation.** Younis et al. [38] detected whether user artifacts are exposed from Web history or email communications in private and non-private modes on four popular mobile browsers. Arshad et al. [12] investigated the performance of the Tor privacy browser for protecting digital browsing traces on Windows 10 and Android 10 devices. Barghouthy et al. [13] proved that critical private data can be found from Orweb (unavailable now) when rooted. Flowers et al. [19] analyzed the private mode of IE, Chrome, Firefox, and Opera. They found that the user's browser evidence was still recoverable in some specific areas. Tsalis et al. [34] found that some worthless files can also recover users' browsing history (e.g., bookmarks) and proposed using a virtual filesystem to protect privacy. Hughes et al. [24] analyzed four mainstream browsers (Brave, Chrome, Edge, and Firefox) and showed that volatile flash memory might disclose critical private data. In our work, we developed an automated analysis framework to detect private browsing trace because it is more suitable for rapidly updated browser apps than the previous manual work.

## 8 CONCLUSION

In this work, we systemically studied mobile private browsing and proposed six rules for browser apps to follow according to the promised capabilities of private browsing from apps and previous works on PC platforms. To verify these rules, we designed an automated device-independent and browser-independent analysis framework, BRODROID. Finally, we implemented BRODROID and conducted experiments based on 49 popular browser apps with more than 5 million downloads. According to the detection reports, BRODROID discovered 58 violations, demonstrating the effectiveness of our tool and revealing the improper implementations of mobile private browsing.

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

## A APPENDIX

The detailed results of evaluation are shown in Table 3. Besides, to better understand the private browsing features of mobile browsers and the goal of private browsing, we collected the private mode capabilities descriptions of our tested browsers from their privacy policy or instructions on the private mode, as shown in Table 4.

**Table 3: Detection results by apps.**

| Package Name | App Version | Chromium-based | Rule1 | Rule2 | Rule3 | Rule4 | Rule5 | Rule6 |
|---|---|---|---|---|---|---|---|---|
| browser4g.fast.internetwebexplorer | 24.10.15 | No | ✓ | ● | ● | ○ | ○ | ● |
| com.aloha.browser | 4.4.4 | Yes | ○ | ○ | ● | ○ | ○ | ○ |
| com.alohamobile.browser | 4.4.4 | Yes | ○ | ○ | ● | ○ | ○ | ○ |
| com.android.chrome | 106.0.5249.118 | Yes | ✓ | ✓ | ✓ | ✓ | ✓ | ○ |
| com.apusapps.browser | 3.1.10 | No | ✓ | ● | ○ | ○ | ● | ● |
| com.brave.browser | 1.45.120 | Yes | ✓ | ○ | ●* | ○ | ○ | ○ |
| com.browser.tssomas | 6.5 | Yes | ✓ | ✓ | ✓ | ✓ | ✓ | ○ |
| com.chrome.beta | 108.0.5359.38 | Yes | ✓ | ✓ | ✓ | ✓ | ✓ | ○ |
| com.chrome.dev | 109.0.5382.0 | Yes | ✓ | ✓ | ✓ | ✓ | ✓ | ○ |
| com.coccoc.trinhduyet | 111.0.174 | Yes | ✓ | ✓ | ✓ | ✓ | ✓ | ○ |
| com.ecosia.android | 7.0.2 | Yes | ✓ | ○ | ●* | ✓ | ○ | ○ |
| com.hsv.freeadblockerbrowser | 96.0.2016123590 | Yes | ✓ | ✓ | ✓ | ✓ | ✓ | ○ |
| com.jio.web | 3.0.6 | Yes | ✓ | ○ | ○ | ○ | ○ | ○ |
| com.kaweapp.webexplorer | 4.5.2 | No | ○ | ● | ● | ○ | ○ | ● |
| com.kiwibrowser.browser | Git210216Gen570536402 | Yes | ○ | ○ | ○ | ○ | ○ | ○ |
| com.mi.globalbrowser | 13.16.1-gn | Yes | ✓ | ✓ | ×* | ✓ | ✓ | ○ |
| com.mi.globalbrowser.mini | 3.9.3 | Yes | ✓ | ● | ● | ○ | ○ | ● |
| com.microsoft.emmx | 105.0.1418.28 | Yes | ✓ | ✓ | ✓ | ✓ | ✓ | ○ |
| com.mx.browser | 6.2.0.1000 | No | ● | ● | ● | ○ | ● | ● |
| com.naver.whale | 2.7.7.2 | Yes | ✓ | ✓ | ✓ | ○ | ○ | ○ |
| com.opera.browser | 65.1.3381.61266 | Yes | ✓ | ○ | ○ | ○ | ○ | ○ |
| com.opera.browser.beta | 72.0.3764.67976 | Yes | ✓ | ○ | ○ | ○ | ○ | ○ |
| com.opera.gx | 1.6.7 | No | ✓ | ○ | ○ | ○ | ○ | ○ |
| com.opera.mini.native | 65.1.2254.63284 | No | ✓ | ○ | ○ | ○ | ○ | ○ |
| com.opera.mini.native.beta | 66.0.2254.63780 | No | ✓ | ○ | ○ | ○ | ○ | ○ |
| com.opera.touch | 2.9.9 | No | ✓ | ○ | ○ | ○ | ○ | ○ |
| com.sec.android.app.sbrowser.beta | 19.0.1.2 | Yes | ● | ○ | ○ | ○ | ○ | ○ |
| com.talpa.hibrowser | v2.5.9.1 | No | ✓ | ○ | ● | ○ | ○ | ○ |
| com.ume.browser.international | 6.0.15 | No | ○ | ● | ● | ○ | ○ | ● |
| com.ume.browser.northamerica | 6.0.15 | No | ○ | ● | ● | ○ | ○ | ● |
| com.xbrowser.play | 3.8.3 | No | ○ | ● | ● | ● | ● | ○ |
| com.yandex.browser | 22.11.0.224 | Yes | × | ○ | ○ | ○ | ○ | ○ |
| fast.explorer.web.browser | 5.9.0 | No | ✓ | × | × | ✓ | ✓ | × |
| mark.via.gp | 4.4.5 | No | ○ | ● | ● | ○ | ○ | ● |
| mobi.mgeek.TunnyBrowser | 12.2.9 | No | ○ | ○ | ○ | ○ | ○ | ● |
| net.fast.web.browser | 5.1.0 | No | ✓ | ● | ● | ○ | ○ | ● |
| org.adblockplus.browser | 3.2.1 | Yes | ✓ | ✓ | ✓ | ✓ | ✓ | ○ |
| org.easyweb.browser | 2.3.0 | No | ✓ | ● | ○ | ○ | ○ | ● |
| org.mozilla.firefox | 105.2.0 | No | ✓ | ○ | ○ | ○ | ○ | ○ |
| org.mozilla.firefox_beta | 106.0b5 | No | ✓ | ○ | ○ | ○ | ○ | ○ |
| privacy.explorer.fast.safe.browser | 2.1.0 | No | ✓ | ● | ● | ○ | ○ | ● |
| webexplorer.amazing.speed | 24.8.14 | No | ✓ | ● | ○ | ○ | ○ | ● |
| com.explore.web.browser † | 3.9.0 | No | ✓ | × | ✓ | ✓ | ✓ | × |
| com.UCMobile.intl † | 13.4.0.1306 | Yes | ✓ | ○ | ● | ● | ○ | ○* |
| cz.seznam.sbrowser † | 9.1.1 | No | ○ | ● | ○ | ● | ● | ● |
| ru.yandex.searchplugin † | 22.97 | Yes | ✓ | ○ | ○ | ○ | ○ | ○ |
| com.duckduckgo.mobile.android ‡ | 5.141.0 | No | ○ | ● | ○ | ○ | ○ | - |
| nu.tommie.inbrowser ‡ | 2.43 | No | ✓ | ○ | ✓ | ○ | ○ | - |
| org.mozilla.focus ‡ | 106.1.0 | No | ○ | ○ | ○ | ○ | ○ | - |

○ The rule is not promised but obeyed;  ● The rule is not promised and violated.  ✓ The rule is promised and obeyed.  × The rule is promised but violated.
*: the results suffer from false negatives or false positives, and we discuss them below.
†: The browsing automation operations for these browsers were assisted with manual work. Our automation strategy cannot deal with them.
‡: These browsers are designed for private browsing without normal mode. Therefore, Rule 6 is not suitable for them.

**Table 4: Promised private browsing capabilities of mobile browsers.**

| Package name | Not promised capabilities | Promised capabilities |
|---|---|---|
| browser4g.fast.internetwebexplorer | N/A | Incognito mode ensures your privacy. Incognito tab won't record any browsing and search history. (Rule 1) |
| com.aloha.browser | Some websites will still be able to track you. | N/A |
| com.alohamobile.browser | Some websites will still be able to track you. | N/A |
| com.android.chrome
com.browser.tssomas
com.chrome.beta
com.chrome.dev
com.coccoc.trinhduyet
com.hsv.freeadblockerbrowser
com.mi.globalbrowser
org.adblockplus.browser | Your activity might still be visible to:
·Website you visit
·Your employer or school
·your internet service provider
Downloads,bookmarks and reading list items will be saved | Browser won't save the following information:
·Your browsing history (Rule 1)
·Cookies and site data (Rule 2-3)
·Information entered in forms (Rule 4)
Other people who use this device won't see your activity
Block third-party cookies (Rule 5) |
| com.apusapps.browser | N/A | Incognito mode ensures your privacy. Incognito tab won't record any browsing and search history. (Rule 1) |
| com.brave.browser | Even though sites you visit in private tabs are not saved locally, they do not make you anonymous or invisible to your ISP, your employer, or to the sites you are visiting. | Sites you visit in private tabs are not saved locally. (Rule 1) |
| com.ecosia.android | N/A | Ecosia won't remember the pages you visited, your search history or your autofill information once you close a tab. (Rule 1 and Rule 4) |
| com.jio.web | Bookmarks added in incognito mode remains private. Your activity might be visible to websites you visit and your internet service provider. | Your search keywords, browsing history are not recorded. (Rule 1) |
| com.kaweapp.webexplorer | N/A | Browse. Erase. Repeat. |
| com.kiwibrowser.browser | N/A | N/A |
| com.mi.globalbrowser.mini | Remenber that the files you download and the bookmarks you add will still be saved. | Your browsing and search history won't be saved in incognito mode. (Rule 1) |
| com.microsoft.emmx | Saves Collections, favorites and download (but not download history).
Does not hide your browsing from your school, employer, or internet service provider.
Does not give you additional protection from tracking by default.
Does not add additional protection to what's available in normal browsing. | Microsoft Edge will delete your browsing history, cookies, and site data, as well as passwords, address, and form data when you close all InPrivate tabs. (Rule 1-4)
Other people using this device won't see your browsing activity.
Prevent Microsoft Bing searches from being associated with you. |
| com.mx.browser | N/A | You're incognito. |
| com.naver.whale | N/A | Incognito mode allows you to browse the web without leaving traces of your Internet connect, including your search history, recent searches, cookies, and temporary files. (Rule 1-3) |
| com.opera.browser | N/A | Opera won't save the browsing history of your private tabs. (Rule 1) |
| com.opera.browser.beta | N/A | Opera beta won't save the browsing history of your private tabs. (Rule 1) |
| com.opera.gx | N/A | Opera GX won't save the browsing history of your private tabs in private mode. (Rule 1) |

| Package name | Not promised capabilities | Promised capabilities |
|---|---|---|
| com.opera.mini.native | N/A | Opera Mini won't save the browsing history of your private tabs. (Rule 1) |
| com.opera.mini.native.beta | N/A | Opera Mini beta won't save the browsing history of your private tabs. (Rule 1) |
| com.opera.touch | N/A | Opera Touch won't save the browsing history of your private tabs in private mode. (Rule 1) |
| com.sec.android.app.sbrowser.beta | N/A | N/A |
| com.talpa.hibrowser | N/A | Hi Browser will not keep your browsing history. (Rule 1) |
| com.ume.browser.international | N/A | N/A |
| com.ume.browser.northamerica | N/A | N/A |
| com.xbrowser.play | N/A | N/A |
| com.yandex.browser | Please note that you will start being "visible" on social media and other sites if you log in. | Your history searches, and passwords will not be saved. (**Rule 1** and Rule 4) |
| fast.explorer.web.browser | N/A | Incognito browsing prevents any information from being stored locally. (Rule 1,**Rule 2,Rule 3**,Rule 4-5,**Rule 6**) |
| mark.via.gp | N/A | You've gone incognito. |
| mobi.mgeek.TunnyBrowser | N/A | N/A |
| net.fast.web.browser | Downloaded files and new bookmarks will still be saved to your device. | We won't remember any history. (Rule 1) |
| org.easyweb.browser | Downloaded files and new bookmarks will still be saved to your device. | We won't remember any history. (Rule 1) |
| org.mozilla.firefox | Does not make you anonymous to websites or your internet service provider make it easier to keep what you do online private from anyone else who uses this device. | Firefox clears your search and browsing history from private tabs when you close them or quit the app. (Rule 1) |
| org.mozilla.firefox_beta | Does not make you anonymous to websites or your internet service provider make it easier to keep what you do online private from anyone else who uses this device. | Firefox Beta clears your search and browsing history from private tabs when you close them or quit the app. (Rule 1) |
| privacy.explorer.fast.safe.browser | Downloaded files and new bookmarks will still be saved to your device. | We won't remember any history. (Rule 1) |
| webexplorer.amazing.speed | N/A | Incognito mode ensures your privacy. Incognito tab won't record any browsing and search history. (Rule 1) |
| com.explore.web.browser | N/A | Incognito browsing prevents any information from being stored locally. (Rule 1,**Rule 2**,Rule 3-5,**Rule 6**) |
| com.UCMobile.intl | Files downloaded and bookmarks will be kept. | Your browsing history and search history won't be recorded. (Rule 1) |
| cz.seznam.sbrowser | N/A | N/A |
| ru.yandex.searchplugin | N/A | Your search and browsing history isn't saved. (Rule 1) |
| com.duckduckgo.mobile.android | N/A | N/A |
| nu.tommie.inbrowser | N/A | You're now leaving InBrowser. All cache and history will be deleted. (Rule 1 and Rule 3) |
| org.mozilla.focus | N/A | N/A |

The **Bold** indicates that the rule was violated.

