# OpenReview forum: "From Promises to Practice: Evaluating the Private Browsing Modes of Android Browser Apps"
_ACM.org/TheWebConf/2024/Conference — TheWebConf24 Oral_

### Official Review · Reviewer_8G4V · 2023-10-28

**Novelty:** 5
**Technical Quality:** 4

**Review:**

## Summary

The work presents a Framework called BroDroid to evaluate the private browsing mode of online browsers based on six different rules. They evaluated their tool on 49 popular mobile online browsers and found that the very popular Chrome browser (63% of global browser market share) fulfilled all privacy criteria tested. Nevertheless, less popular ones like Samsung Internet Beta (#3 in global browser share [for Samsung Internet], ~4%), UC Browser (#5, 1.6%) and Yandex Browser (unknown rank) violated some of the privacy rules.

## Pros

- Protecting the privacy of online users is a very relevant topic, and research should evaluate whether online browser keep their communicated privacy promised. Therefore, this work provides valuable insights into the state of practice.

- Tool and methodology allows to test privacy properties Android online browser apps at scale.

## Cons

- Impact of work rather unclear. The paper does not state how many potential users have been affected by the privacy violations. I think it could be very helpful to see some hard facts, e.g., in Table 1. Also, I had the impression while reading that the most popular online browsers seem to protect the privacy of their users, which to me is a good finding. However, the paper seems to focus more on the negative findings, which do not have "that much" impact (< 5-10% of mobile browser share, which should still be hundreds of millions of people). By just stating the absolute number ("58 violations") without setting them into context (i.e., how many browsers were affected), it appears to me like an attempt to oversell the results (although the findings are still valuable).

- Motivation partly unclear.
    - The outlined test criteria appear not to be justified to me. I do not see information on why it was decided to check the internal storage. Was this based on previous related work or on the authors previous experience? And by which criteria were the "several popular web browsers" selected?
    - Also, the private rules in Section 3 are all based on previous research on PC browsers. Why was research on Android devices not considered (like Kywe wt al. [11])? Also, the threat models seem to be taken from Aggarwal et al. [11], but this was not fully stated in the text.

- Limitations of approach not completely discussed. It seems like study was used in a local environment without HTTPS, which is currently the standard for web browsers. Can this have an impact on the study outcome, e.g., browsers never store or submit passwords because the connection is not secure?

- Writing is good, but still can be improved.
    - Which browsers were tested? I see that they are in the Appendix, but they were never stated in the main part of the paper. Also, why is the Samsung Internet Beta not mentioned in Table 4?
    - The automated testing was only possible by changing the FLAG_SECURE parameter. Can you please explain what this parameter does? Did it have any impact on the study outcome?
    - The naming of the different rules to "Rule 1", "Rule 2" etc. seems to be confusing for me as a reader. I have to look up the rules pages before that to understand what each rule was about. Therefore, maybe it is helpful to give each rule an abbreviation to help me remember the rules (e.g., No-Browser-History, No-Cookies, No-Web-Cache, No-Forms, Block-Third-Parties, No-Cookie-Sharing).

## Conclusion

I really like the work, and would lean towards it. Nevertheless, there are multiple points that have to be improved in the paper before it could be in a publishing-ready state. Therefore, I would reject the paper in its current form but encourage the authors to submit this research in improved form.

## Further Notes

- Will BroDroid be provided as open source software to the community? I think that it could be very valuable for both browser developers, society, and researchers aiming to replicate your research at a later point in time (e.g., did browsers really fix the issues reported?).

- Can the tool be used on an emulated browser as well?

- 2.2, local adversary: What about state actors that could be able to install malware like Pegasus?

## Post-Rebuttal

Thanks for responding in the rebuttal. Nevertheless, for a top-tier conference like WWW, the "low" market share of the browsers affected by the presented attack might be an issue. Therefore, the paper in its current form would be a borderline case for me.

**Questions:**

See Review.

**Ethics Review Description:**

-

**Reviewer Confidence:**

3: The reviewer is confident but not certain that the evaluation is correct

**Scope:**

4: The work is relevant to the Web and to the track, and is of broad interest to the community

---

### Official Review · Reviewer_LaeC · 2023-11-17

**Novelty:** 2
**Technical Quality:** 2

**Review:**

The main contribution of this paper is the creation of a software framework called BroDroid that is able to turn on private browsing, access a certain website, simulate a few actions (repeated clicks of the back button), and show that some information is being leaked. Given that no explicit taint propagation or value inspection is being done, the methods to detect leaks are somewhat simple such as measuring the size of files created on storage media, or check if a network packet contains a cookie. The software requires a rooted mobile phone and is quite restrictive. BroDroid itself runs on a desktop/laptop CPU.

The fact is that in the light of related work, a browser automation framework to simulate actions in the private/incognito mode is not very novel. The related work section lists a few competing works.

Moreover, the mechanisms for detecting if a site is leaking data or not are very basic and so are the six rules. There is a need to seriously overhaul the paper in my humble opinion.

**Questions:**

1. Have you considered fingerprinting?
2. Have you considered more subtle forms of data leakage such as via side channels?
3. Have you considered ML-based approaches to figure out the method to enter the private browsing mode?

**Ethics Review Description:**

It is fine by me.

**Reviewer Confidence:**

3: The reviewer is confident but not certain that the evaluation is correct

**Scope:**

3: The work is somewhat relevant to the Web and to the track, and is of narrow interest to a sub-community

---

### Official Review · Reviewer_87HW · 2023-11-22

**Novelty:** 4
**Technical Quality:** 6

**Review:**

The paper presents a study of private browsing in 54 popular web browsers
available in Android.  Private browsing is a mode available in most web
browsers that allows the user to bootstrap a clean session, with no use of
cookies already stored in the browser, while, once the session is finished, the
environment (cookies, history) is, again, wiped. Since there is no standard of
private browsing, each vendor implements the mode on their own. Several studies
exist so far for desktop browsers, however, limited research has been done in
mobile browsers.

(+)

- Surprisingly, the paper highlights that there is a plethora of web browsers
  for mobile devices, that are very inconsistent with each other, as far as
  private browsing is concerned.

- The tool for automatically inferring violations deals with some interesting
  challenges.

(-)

- Weak comparison with desktop web browsers.

- Unclear if the violations found are complete.

Details

Comparison with desktop web browsers. The difference between web browsers for
PCs and mobile platforms is not clearly highlighted. For instance, there is an
argument related to the different file-accessing policies of mobile apps
compared to desktop apps, however the connection of this and private browsing
is unclear. Furthermore, in the motivation, the difference between the two
platforms is based on the fact that saved files are protected by the mobile
operating system (e.g., Android), and thus the "internal storage" should be
more carefully explored for leftover traces. I believe this is true for both
desktop and mobile web browsers, since "internal storage" may mean several
things for both platforms.

Violations. I find very interesting the part with the browser violations.
First, the rules as discussed in Section 3 make sense and they are in
accordance to several related papers. Second, it is interesting that most
mobile browsers violate some of the rules. Nevertheless, I have some
reservations related to the soundness of thee reported violations. For
instance, in the motivating example of Sec 2.1, with Yandex browser, it is said
that some browsing information is stored in a file (in
com.yandex.browser/databases/dashboard_service), however it is not clear for
how long this information is stored or if it is actually reused in future
sessions.

In general, there is a fundamental issue with the analysis time (i.e., how long
private information is kept in the browser environment) which appears in the
entire experimental process. The authors chose a threshold of 30 seconds, after
closing browsers, to infer if information is still not cleaned and they admit
that this threshold may influence the evaluation results ("In addition, we
found that the storage of private browsing data is related to time for some
browsers. That is, the time to analyze storage after closing private browsing
can affect the evaluation results."). I would therefore, suggest, the following.

(a) Split violations to two sets, one that is affected by this 30 seconds
threshold, and one that is not.

(b) For the set that is affected, perform the analysis for various thresholds
(let's say, 30 seconds, 1 minute, 5 minutes).

(c) Possibly, for the set that is affected, have a second analysis pass where
the browser is re-launched (information may be cleaned on this step).

**Questions:**

- What is the difference of "internal storage" between desktop and mobile web
  browsers?

- How was the 30-seconds threshold selected in your analysis?

- Can you perform a similar analysis with a longer threshold, let's say a few
  minutes?

**Reviewer Confidence:**

3: The reviewer is confident but not certain that the evaluation is correct

**Scope:**

4: The work is relevant to the Web and to the track, and is of broad interest to the community

---

### Official Review · Reviewer_6LHq · 2023-11-24

**Novelty:** 5
**Technical Quality:** 5

**Review:**

This paper studies the behavior of private browsing modes on mobile browsers on the Android platform. In particular, the authors summarized the expected privacy rules for mobile browsers to follow (based on the prior works, privacy expectations and stated private mode descriptions) and tested those with an automated analysis framework, BroDroid. Overall, they analyzed 49 popular mobile browsers, discovering 58 violations.

The paper provides interesting technical findings and highlights the necessity to improve privacy practices of mobile browsers. In addition, the authors find such interesting trends as the fact that Chromium-based browsers may introduce privacy issues while implementing customization. I also like that the authors reported their discoveries to browser vendors and stated that some of them already acknowledged their findings.

The main weakness of the paper is the clarity of the threat model. In general, browser private modes have a history of confusion regarding privacy expectations, and I believe that the six privacy rules proposed by the authors are more driven by their ultimate privacy expectations rather than the promised capabilities by vendors. Several things that must be clarified:
- Incognito and private modes are often used to avoid saving the browsing history of a user’s account by browser vendor on the server side, however it looks like the BroDroid evaluates only the local database.  Have you tested any signs of server-side storage of browsing history?
- Some of the promised private browsing capabilities listed in Table 4 are rather vague, such as  “won’t save”, “won’t be saved”. For example, “your browsing and search history won’t be saved in incognito mode” - did you consider it as saved locally or on the server side?
- “We believe that browser app developers should not require users to actively turn off privacy mode before exiting the browser” - does any browser explicitly state on how fast they clean the browsing history after the browser is closed? Is it different from the behavior of the desktop browsers? It would be useful to repeat experiments with a longer waiting time or checking local storage after a restart, etc.
- The authors state that “due to the enormous differences between PC and mobile platforms, PC browser and mobile browsers face different kind of adversaries and the security mechanisms of PC private browsing can not be applied to mobile browser” - I found this claim not really elaborated in the paper, moreover it explains that mobile browsers have sandboxing and so are more secure. Could you explain the different adversaries? Also, it would be useful to compare results with desktop browsers of the correspondent vendors.

It would be useful to list all the violated promised capabilities in the paper in addition to the summarized rules, which will give a better understanding on the discovered issues.

In addition, I have some questions on the technical implementation of the BroDroid:
- Why test over websites on local LAN? Was there a specific reason for it? I wonder if browser behavior may differ for such local websites.
- How technically challenging it would be to extend the framework for iOS browsers?
- How much manual effort does it still require for such research? It looks like not all steps were fully automated.

**Questions:**

- Incognito and private modes are often used to avoid saving the browsing history of a user’s account by browser vendor on the server side, however it looks like the BroDroid evaluates only the local database. Have you tested any signs of server-side storage of browsing history?
- Why test over websites on local LAN? Was there a specific reason for it? I wonder if browser behavior may differ for such local websites.

**Reviewer Confidence:**

4: The reviewer is certain that the evaluation is correct and very familiar with the relevant literature

**Scope:**

4: The work is relevant to the Web and to the track, and is of broad interest to the community

---

### Decision · Program_Chairs · 2024-01-22

**Decision:**

Accept (Oral)

**Comment:**

This paper presents a study on the implementation of private browsing modes in Android mobile browsers, from a digital forensics perspective. Since no standard exists to define the behavior of private browsing modes, their implementation can vary widely across different browsers. This study aims to uncover whether any private browsing information is left behind in the device's storage. To that end, the authors present BroDroid, a tool that evaluates private browsing mode implementations based on six different rules.

 During the review process, the reviewers agreed about the paper's novelty and impact, as the proposed tool and methodology enable the testing of privacy properties in Android browser apps at scale. The reviewers also appreciated the reported findings, as multiple inconsistencies were found across different browsers. The tool will also be open-sourced, and can enable additional security research in the future.

 At the same time, the reviewers voiced their concerns about certain methodological issues and potential implications that they may have on the study's findings. This included the fact that websites were tested on a local LAN, and that the threshold for assessing private data being stored is low.

 ---